# Randomized controlled trial: Quantifying the impact of disclosing uncertainty on adherence to hypothetical health recommendations

**Hannah Mendoza****, Lucy D'Agostino McGowan**  *

Department Statistical Sciences, Wake Forest University, Winston-Salem, NC, United States of America

* mcgowald@wfu.edu

## Abstract

We conducted a randomized controlled trial to assess whether disclosing elements of uncertainty in an initial public health statement will change the likelihood that participants will accept new, different advice that arises as more evidence is uncovered. Proportional odds models were fit, stratified by the baseline likelihood to agree with the final advice. 298 participants were randomized to the treatment arm and 298 in the control arm. Among participants who were more likely to agree with the final recommendation at baseline, those who were initially shown uncertainty had a 46% lower odds of being more likely to agree with the final recommendation compared to those who were not (OR: 0.54, 95% CI: 0.27-1.03). Among participants who were less likely to agree with the final recommendation at baseline, those who were initially shown uncertainty have 1.61 times the odds of being more likely to agree with the final recommendation compared to those who were not (OR: 1.61, 95% CI: 1.15-2.25). This has implications for public health leaders when assessing how to communicate a recommendation, suggesting communicating uncertainty influences whether someone will adhere to a future recommendation.

**Data Availability Statement:** All data and code to reproduce the analyses found here can be found here: https://github.com/LucyMcGowan/plos-rct-quantifying-impact-of-disclosing-uncertainty.

## Introduction

Recent leadership decisions surrounding the COVID-19 pandemic have brought into question how decisions and guidelines regarding science and public health are communicated to the public [1–4]. It has been demonstrated that evidence-based information increases the self-reported likeliness to adopt a health recommendation [4]. There remains debate, however, over how disclosing elements of uncertainty in these recommendations influences the public's likelihood to accept them. Research and discussion such as that in "Communicating scientific uncertainty" offer guidelines regarding *how* uncertainty should be communicated to best serve decision makers [5]. However, there is a lack of evidence surrounding *whether* communicating this information actually influences what an individual decides to do in the context of health decision making. A recent review by Van der Bles et al. (2019) suggested that while there have been limited studies that have examined the impact of uncertainty on decision making on topics such as financial decisions [6, 7] and weather [8, 9], as well as studies that demonstrate the

**Funding:** The author(s) received no specific funding for this work.

**Competing interests:** The authors have declared that no competing interests exist.

impact of communicating uncertainty on public trust in health contexts [10, 11], there have not been studies that specifically examine the impact of uncertainty on health decision making [12]. We aim to specifically examine decisions in matters of public health and whether there is a difference in likelihood of accepting changing advice if uncertainty is communicated along the way, conditional on the decision maker's likelihood of acceptance pre-communication.

## Methods

### Study overview

This randomized study was conducted using Qualtrics (Qualtrics, Provo, UT). Participants were asked, "How likely are you to sanitize your mobile phone daily?" and given five available options, from "Not at all likely" to "Very likely". Additionally, demographic information (age, gender, race, ethnicity, educational attainment, and income) was collected. Participants were randomized to see a public health recommendation without uncertainty (Statement 1A) and with uncertainty (Statement 1B). We refer to the treatment arm as those randomized to the 'uncertainty' group and the control arm as those randomized to the 'no uncertainty' group.

**Statement 1A (Control arm)**. There is speculation that regularly sanitizing mobile phones can reduce the number of illnesses acquired. However, based on current evidence, public health officials do not recommend that people sanitize their mobile phones daily.

**Statement 1B (Treatment arm)**. There is speculation that regularly sanitizing mobile phones can reduce the number of illnesses acquired. A study tested whether sanitizing phones yielded a significant difference in the number of illnesses acquired. The study did not find that sanitizing phones made a difference. The study was performed only on landline phones. Based on current evidence, public health officials do not recommend that people sanitize their mobile phones daily.

They were then asked, "How likely are you to accept this advice from public health officials to not sanitize your mobile phone daily?" and given five available options, from "Not at all likely" to "Very likely".

Then, both groups were shown a new public health recommendation (Statement 2).

**Statement 2 (All participants)**. Additional testing was done on mobile phone sanitation, and found there were significantly fewer illnesses acquired among people who sanitized their mobile phones compared to those who did not. Based on this new evidence, public health officials recommend that people sanitize their mobile phones daily.

They were then asked, "How likely are you to accept this advice from public health officials to sanitize your mobile phone daily?" and given five available options, from "Not at all likely" to "Very likely".

This study was approved by Wake Forest University's IRB (#eIRB00023970).

### Participant recruitment and eligibility

Participants were recruited using Prolific (www.prolific.co) [November 2020]. Prolific is an online service that connects researchers with research participants based on specified eligibility criteria. A total of 603 participants were recruited and randomly assigned to see the initial public health recommendation with or without uncertainty. Eligibility criteria included residing in the United States and being 18 years or older.

### Analysis

Our primary analysis examined the relationship between whether the participant was initially communicated uncertainty in Statement 1 and their likelihood to accept the new advice in

Statement 2, conditional on their pre-treatment likelihood to agree with this recommendation (to sanitize their mobile phone). To do so, we first fit a proportional odds model [13] estimating their likelihood to accept the public health recommendation in Statement 2 with an interaction between treatment group and self-reported likelihood to agree at baseline (determined as "likely" if ranked 4 or 5). If the interaction was significant at the $\alpha = 0.05$ level, we examined the same model stratified by baseline likelihood to agree. 95% confidence intervals are reported and significance was determined using an $\alpha = 0.05$ cutoff. Sensitivity analyses were performed, examining the model stratified by the 5-level response to participants' baseline likelihood to agree rather than dichotomizing this. The proportional odds assumption was tested using the Brant test [14].

As a secondary analysis, we examined the impact of the uncertainty communication on their acceptance of the *initial* recommendation (after Statement 1). We fit a proportional odds model estimating participants' likelihood to accept the public health recommendation in Statement 1 with an interaction between treatment group and self-reported likelihood to agree at baseline. If the interaction was significant at the $\alpha = 0.05$ level, we examined the same model stratified by baseline likelihood to agree.

## Results

603 participants were randomized. Six participants did not answer primary analysis question and one did not meet eligibility criteria (self reported age was 17), and therefore were removed prior to analysis, resulting in $n = 596$ observations with 298 randomized to the treatment arm ('uncertainty' in the initial public health recommendation) and 298 in the control arm ('no uncertainty' in the initial public health recommendation). We examined baseline characteristics between treatment arms (Table 1). The median age in the sample was 29 (IQR: 22-36), 51% were female. At baseline, 26% reported that their likelihood of sanitizing their mobile phone daily was a 4-5 on a 5 point scale, with 1 indicating "not at all likely" and 5 indicating "very likely". We categorize those that indicated a 4-5 at baseline as "more likely to agree with the final recommendation at baseline", and those that indicated a 1-3 at baseline as "less likely to agree with the final recommendation at baseline".

### Primary analysis

All analyses were conducted using R version 4.1.2 [15–19]. Fig 1 shows the distribution of acceptance of the final recommendation (Statement 2). The top panel shows the acceptance between treatment arms among those who were more likely to agree with the final recommendation at baseline; the bottom panel shows acceptance between treatment arms among those less likely to agree with the final recommendation at baseline. Among those who were more likely to agree with the final recommendation at baseline, 82.1% of those shown the initial recommendation with 'uncertainty' reported that they were likely or very likely to comply. This is a decrease from the control arm that was shown the initial recommendation with 'no uncertainty', of whom 93.4% reported being likely or very likely to comply (Fig 1A). Recall that at baseline 100% of this subgroup indicated that they would sanitize their phone (the final recommendation). This suggests that expressing uncertainty in the initial recommendation to this subgroup led to an absolute decrease in adherence to the final recommendation of 11.3%, or a relative percent change of -12.1%. We see the opposite effect among those who were less likely to agree with the final recommendation at baseline. In this subgroup, 66.8% of those shown the initial recommendation with 'uncertainty' reported that they were likely or very likely to comply. This is an increase compared to the control arm that was shown the initial recommendation with 'no uncertainty', of whom 55% reported being likely or very likely to comply

**Table 1. Baseline characteristics.**

| Characteristic | No Uncertainty, N = 298 | Uncertainty, N = 298 |
|---|---|---|
| Age | 29 (18, 22, 36, 78) | 28 (18, 23, 36, 75) |
| Gender | | |
| Female | 148 (50%) | 159 (53%) |
| Male | 147 (49%) | 127 (43%) |
| Prefer to self describe | 3 (1.0%) | 12 (4.0%) |
| Race | | |
| American Indian/Alaska Native | 1 (0.3%) | 4 (1.3%) |
| Asian | 46 (15%) | 40 (13%) |
| Black or African American | 22 (7.4%) | 24 (8.1%) |
| White | 217 (73%) | 214 (72%) |
| Prefer to self describe | 9 (3.0%) | 11 (3.7%) |
| Prefer not to answer | 3 (1.0%) | 5 (1.7%) |
| Ethnicity | | |
| Hispanic or Latino | 27 (9.1%) | 31 (10%) |
| Not Hispanic or Latino | 266 (89%) | 262 (88%) |
| Prefer not to answer | 5 (1.7%) | 5 (1.7%) |
| Educational Attainment | | |
| Did not attend high school | 1 (0.3%) | 1 (0.3%) |
| Some high school | 3 (1.0%) | 4 (1.3%) |
| High school graduate | 38 (13%) | 44 (15%) |
| Some college | 107 (36%) | 93 (31%) |
| College graduate | 87 (29%) | 99 (33%) |
| Some postgraduate work | 7 (2.3%) | 14 (4.7%) |
| Postgraduate degree | 55 (18%) | 43 (14%) |
| Average Household Income | | |
| $0-$24,999 | 56 (19%) | 49 (16%) |
| $25,000-$49,999 | 68 (23%) | 73 (24%) |
| $50,000-$74,999 | 68 (23%) | 68 (23%) |
| $75,000-$99,999 | 41 (14%) | 31 (10%) |
| $100,000-$124,999 | 19 (6.4%) | 32 (11%) |
| $125,000-$149,999 | 11 (3.7%) | 15 (5.0%) |
| $150,000-$174,999 | 9 (3.0%) | 13 (4.4%) |
| $175,000-$199,999 | 3 (1.0%) | 2 (0.7%) |
| $200,000 and up | 14 (4.7%) | 9 (3.0%) |
| Prefer not to answer | 9 (3.0%) | 6 (2.0%) |
| (baseline) How likely are you to sanitize your phone? | | |
| 1 Not at all likely | 91 (31%) | 89 (30%) |
| 2 | 91 (31%) | 85 (29%) |
| 3 | 40 (13%) | 46 (15%) |
| 4 | 43 (14%) | 40 (13%) |
| 5 Very likely | 33 (11%) | 38 (13%) |

[1] Median (Minimum, IQR, Maximum); n (%)

(Fig 1B). Recall that at baseline 0% of this subgroup indicated that they would sanitize their phone (the final recommendation). This suggests that expressing uncertainty in the initial recommendation to this subgroup led to an absolute increase in adherence to the final recommendation of 11.8%, or a relative percent change of 21.5%.

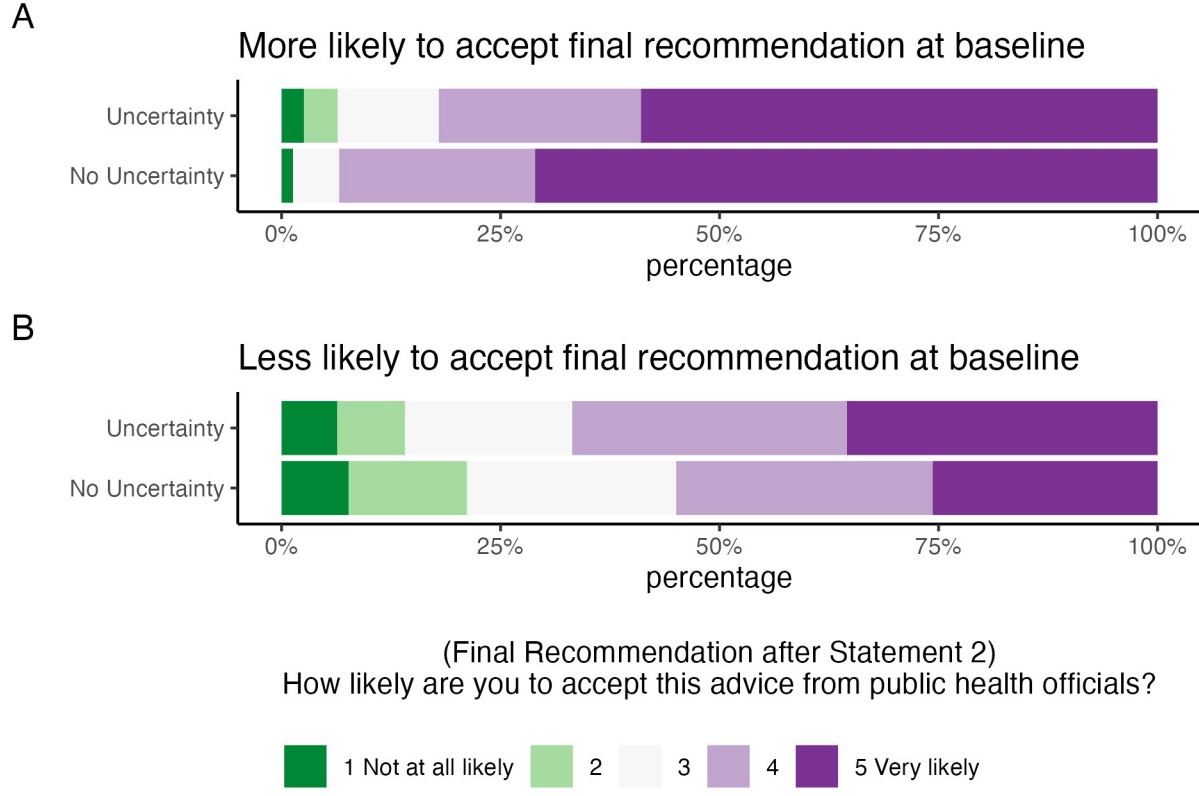

**Fig 1. Distribution of agreement with the final recommendation.** The top panel (A) shows the agreement between treatment arms among those who were more likely to agree with the final recommendation at baseline; the bottom panel (B) shows agreement between treatment arms among those less likely to agree with the final recommendation at baseline.

A proportional odds model was fit, estimating the reported likelihood of accepting the final recommendation (from 1 "not at all likely" to 5 "very likely") based on the treatment group, whether the participant was likely to agree with the final recommendation at baseline, and an interaction between the two. The p-value for the interaction term was significant (p = 0.005). We then examined two proportional odds models, stratified by whether the participant was likely to agree with the final recommendation at baseline (Table 2). Our model fit among those who were more likely to agree with the final recommendation at baseline has an odds ratio of 0.54 (95% CI: 0.27, 1.03), suggesting that participants who were initially shown uncertainty have a 46% lower odds of being more likely to agree with the final recommendation compared to those who were not. Our model fit among those who were less likely to agree with the final recommendation at baseline has an odds ratio of 1.61 (95% CI: 1.15, 2.25), suggesting that participants who were initially shown uncertainty have 1.61 times the odds of being more likely to agree with the final recommendation compared to those who were not. Neither model showed significant deviation from the proportional odds assumption, as tested by the Brant test (p = 0.2; p = 0.85).

## Sensitivity analyses

For the main analysis, we considered participants more likely to agree with the final recommendation at baseline if they answered the question "How likely are you to sanitize your mobile phone daily?" with a 4 or 5 (on a scale from "1: Not at all likely" to "5: Very likely"). Likewise, we considered participants less likely to agree with the final recommendation at

**Table 2. Proportional odds model examining the impact of seeing uncertainty in the initial recommendation on accepting a future recommendation.**

| Group | Characteristic | OR | 95% CI |
|---|---|---|---|
| Model fit on the whole sample with interaction | Treatment | | |
| | No Uncertainty | — | — |
| | Uncertainty | 1.61 | 1.15, 2.26 |
| | Baseline Agreement | | |
| | no | — | — |
| | yes | 7.60 | 4.45, 13.4 |
| | Treatment * Baseline Agreement | | |
| | Uncertainty * yes | 0.34 | 0.16, 0.70 |
| Model among baseline agreers | Treatment | | |
| | No Uncertainty | — | — |
| | Uncertainty | 0.54 | 0.27, 1.03 |
| Model among baseline disagreers | Treatment | | |
| | No Uncertainty | — | — |
| | Uncertainty | 1.61 | 1.15, 2.25 |

OR = Odds Ratio, CI = Confidence Interval

baseline if they responded with 1, 2, or 3. Here, we examine how sensitive the results are to this cutoff. Table 3 shows the model stratified by the participants' response without dichotomizing, resulting in 5 separate proportional odds models fit. The point estimates of the results appear consistent with the main result. The models where the participant responded that they were less likely to sanitize their phone at baseline (indicated by a response of a 1, 2, or 3) have odds ratios greater than one, indicating that seeing uncertainty on the initial recommendation may increase the likelihood of adherence to a subsequent recommendation. The models where the

**Table 3. Sensitivity analysis: Proportional odds model fit stratified by baseline likelihood to sanitize mobile phone (with 5 levels).**

| Group | Characteristic | OR | 95% CI |
|---|---|---|---|
| Baseline likelihood to sanitize mobile phone = 1: Not at all likely | Treatment | | |
| | No Uncertainty | — | — |
| | Uncertainty | 1.42 | 0.85, 2.40 |
| Baseline likelihood to sanitize mobile phone = 2 | Treatment | | |
| | No Uncertainty | — | — |
| | Uncertainty | 1.76 | 1.03, 3.04 |
| Baseline likelihood to sanitize mobile phone = 3 | Treatment | | |
| | No Uncertainty | — | — |
| | Uncertainty | 1.91 | 0.85, 4.32 |
| Baseline likelihood to sanitize mobile phone = 4 | Treatment | | |
| | No Uncertainty | — | — |
| | Uncertainty | 0.47 | 0.20, 1.08 |
| Baseline likelihood to sanitize mobile phone = 5: Very likely | Treatment | | |
| | No Uncertainty | — | — |
| | Uncertainty | 0.53 | 0.16, 1.58 |

OR = Odds Ratio, CI = Confidence Interval

**Table 4. Proportional odds model examining the impact of seeing uncertainty in the initial recommendation on accepting that recommendation.**

| Group | Characteristic | OR | 95% CI |
|---|---|---|---|
| Model fit with interaction | Treatment | | |
| | No Uncertainty | — | — |
| | Uncertainty | 0.64 | 0.46, 0.90 |
| | Baseline Agreement | | |
| | no | — | — |
| | yes | 0.40 | 0.25, 0.63 |
| | Treatment * Baseline Agreement | | |
| | Uncertainty * yes | 0.95 | 0.49, 1.82 |
| Model fit without interaction | Treatment | | |
| | No Uncertainty | — | — |
| | Uncertainty | 0.63 | 0.47, 0.84 |

OR = Odds Ratio, CI = Confidence Interval

participant responded that they were more likely to sanitize their phone at baseline (indicated by a response of a 4 or 5) have odds ratios less than one, indicating that seeing uncertainty on the initial recommendation may decrease the likelihood of adherence to a subsequent recommendation.

## Secondary analysis

As a secondary analysis, we examined the impact of seeing uncertainty in the initial recommendation on the likelihood to accept *that* recommendation. A proportional odds model was fit, estimating the reported likelihood to follow the *initial* recommendation (from 1 "not at all likely" to 5 "very likely") based on the treatment group, whether the participant was likely to agree with the final recommendation at baseline, and an interaction between the two. The p-value for the interaction term was not significant (p = 0.872). Therefore, we removed this from the model and examined a single proportional odds model among all participants, examining the overall impact of uncertainty on the likelihood to accept the initial recommendation. This model has an odds ratio of 0.63 (95% CI: 0.47, 0.84), suggesting that participants who were shown uncertainty have a 37% lower odds of being more likely to agree with the initial recommendation compared to those who were not (Table 4). This model did not show significant deviation from the proportional odds assumption, tested via the Brant test (p = 0.87).

## Conclusion

In this randomized controlled trial we observed that disclosing elements of uncertainty in an initial public health statement will change the likelihood that participants will accept new, different advice that arises as more evidence is uncovered. The direction of this effect differed based on whether the participants indicated that they would agree with the final recommendation at baseline. Specifically, among participants who were more likely to agree with the final recommendation at baseline, those who were initially shown uncertainty were less likely to agree with the final recommendation compared to those who were not. Conversely, among participants who were less likely to agree with the final recommendation at baseline, those who were initially shown uncertainty were more likely to agree with the final recommendation compared to those who were not.

## Discussion

These results suggest that presenting public health recommendations with uncertainty can both increase and decrease adherence to future recommendations in the presence of changing evidence, depending on the individuals' baseline likelihood to do what is being recommended prior to seeing the recommendation. This may have implications for how recommendations are communicated. For example, if it is known a priori that the majority of the public would not have elected to take a certain action prior to it being recommended, then presenting the recommendation with proper uncertainty may have a larger positive impact on future adherence to recommendations, particularly if they have some potential to change. The secondary results show, however, that in the presence of a *single* recommendation, presentation with uncertainty may decrease the acceptance of the recommendation. The decision on the method of communication may depend on the prevalence of the public health action prior to the recommendation being made as well as the probability that the recommendation might change in the future. Many public health recommendations, particularly those being made during an ever changing pandemic, have a high probability of changing in the future, making this result highly relevant. For example, the recommendation to wear masks will come and go depending on the prevalence and impact of the infectious disease of concern in the community. When *initially* communicating this new recommendation to the public, an explanation of the reasoning for the recommendation as well as any potential uncertainty may have resulted in increased adherence among those who previously would not have been likely to wear a mask in the recommended settings prior to the recommendation taking place. For example, a survey conducted on March 12-14, 2020 found that only 12% of respondents from the United States reported wearing a face mask to protect themselves from COVID-19 [20]. At the time, the public health recommendation from the CDC was: "If you are NOT sick: You do not need to wear a face mask unless you are caring for someone who is sick (and they are not able to wear a face mask)" [21]. Based on the results from our study, if the initial recommendation to avoid wearing face masks had been presented with the uncertainty communicated, we might expect some of this 12% to be less likely to follow the recommendation. However, 88% were *not* likely to wear masks a priori; our results suggest this population may have ultimately had higher compliance with the subsequent recommendation to wear face masks had the initial communication been made with uncertainty.

This study has several limitations. First, the sample was predominantly younger; future work will explore whether this generalizes to an older population. Second, the study was purely hypothetical; we were not able to observe whether participants actually complied with given recommendations, but rather how they self-reported that they *would* comply. Further study is needed to examine whether this self-reported difference holds in actual decision making. Finally, while we attempted to create a hypothetical situation that mimicked a public health scenario, we do not know with certainty that these results would translate should the scenario change.

Our results have the potential to impact the methods used to communicate public health recommendations, suggesting that consideration for base-line adherence may drive how information should be communicated to maximize adherence.

## Author Contributions

**Conceptualization:** Lucy D'Agostino McGowan.

**Data curation:** Lucy D'Agostino McGowan.

**Formal analysis:** Hannah Mendoza, Lucy D'Agostino McGowan.

**Investigation:** Lucy D'Agostino McGowan.

**Methodology:** Lucy D'Agostino McGowan.

**Project administration:** Lucy D'Agostino McGowan.

**Resources:** Lucy D'Agostino McGowan.

**Software:** Hannah Mendoza, Lucy D'Agostino McGowan.

**Supervision:** Lucy D'Agostino McGowan.

**Validation:** Lucy D'Agostino McGowan.

**Visualization:** Lucy D'Agostino McGowan.

**Writing – original draft:** Hannah Mendoza, Lucy D'Agostino McGowan.

**Writing – review & editing:** Lucy D'Agostino McGowan.

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
