## [Decision Letter · Decision Letter 0]

13 Jun 2022

PONE-D-22-13360Quantifying the Impact of Disclosing Uncertainty on Health Decision MakingPLOS ONE

Dear Dr. D'Agostino McGowan,

Thank you for submitting your manuscript to PLOS ONE. After careful consideration, we feel that it has merit but does not fully meet PLOS ONE’s publication criteria as it currently stands. Therefore, we invite you to submit a revised version of the manuscript that addresses the points raised during the review process.

We look forward to receiving your revised manuscript.

Kind regards,

Felix Bongomin, MB ChB, MSc, MMed, FECMM

Academic Editor

PLOS ONE

Journal Requirements:

Additional Editor Comments:

Please address the reviewers comments

Reviewers' comments:

Reviewer's Responses to Questions

**Comments to the Author**

1. Is the manuscript technically sound, and do the data support the conclusions?

Reviewer #1: Partly

2. Has the statistical analysis been performed appropriately and rigorously? 

Reviewer #1: Yes

3. Have the authors made all data underlying the findings in their manuscript fully available?

Reviewer #1: Yes

4. Is the manuscript presented in an intelligible fashion and written in standard English?

Reviewer #1: Yes

5. Review Comments to the Author

Reviewer #1: 1. The title of this paper is not appropriate for the content. Consider revising it by colouring it with some specific elements of the research work

2. There should be a brief description of how the recruitment of participants was conducted – reproducible explanation.

3. Conclusion in the abstract should more precise. The present conclusion is vague.

4. The conclusion in the body of the paper was not clearly indicated

5. I consider some of the authorities cited in this work as archaic. It may be more appropriate to cite more current authorities (if available) for issues that are rapidly changing

6. PLOS authors have the option to publish the peer review history of their article (what does this mean?). If published, this will include your full peer review and any attached files.

Reviewer #1: **Yes: **Victor Eneojo (VE) ADAMU

---

## [Author Response · Author response to Decision Letter 0]

16 Sep 2022

Thank you for your helpful comments, please find a point by point response below.

1. The title of this paper is not appropriate for the content. Consider revising it by colouring it with some specific elements of the research work

*We have updated the title to read: Randomized Controlled Trial: Quantifying the Impact of Disclosing Uncertainty on Adherence to Hypothetical Health Recommendations*

2. There should be a brief description of how the recruitment of participants was conducted – reproducible explanation.

*The section "Participant recruitment and eligibility" covers this topic. We have added details to specify what recruitment via Prolific entails.*

3. Conclusion in the abstract should more precise. The present conclusion is vague.

*We have updated the conclusion in the abstract to be more specific*

4. The conclusion in the body of the paper was not clearly indicated

*We have added a conclusion section that reads:*

*In this randomized controlled trial we observed that disclosing elements of uncertainty in an initial public health statement will change the likelihood that participants will accept new, different advice that arises as more evidence is uncovered. The direction of this effect differed based on whether the participants indicated that they would agree with the final recommendation at baseline. Specifically, among participants who were more likely to agree with the final recommendation at baseline, those who were shown uncertainty initially were less likely to agree with the final recommendation compared to those who were not. Conversely, among participants who were less likely to agree with the final recommendation at baseline, those who were shown uncertainty initially were more likely to agree with the final recommendation compared to those who were not.*

5. I consider some of the authorities cited in this work as archaic. It may be more appropriate to cite more current authorities (if available) for issues that are rapidly changing

*Thank you, we have added several citations to the introduction in an effort to cite current authorities on the topic.*

---

## [Decision Letter · Decision Letter 1]

14 Nov 2022

Randomized Controlled Trial: Quantifying the Impact of Disclosing Uncertainty on Adherence to Hypothetical Health Recommendations

PONE-D-22-13360R1

Dear Dr. D'Agostino McGowan,

We’re pleased to inform you that your manuscript has been judged scientifically suitable for publication and will be formally accepted for publication once it meets all outstanding technical requirements.

Kind regards,

Felix Bongomin, MB ChB, MSc, MMed, FECMM

Academic Editor

PLOS ONE

Additional Editor Comments (optional):

Reviewers' comments:

Reviewer's Responses to Questions

**Comments to the Author**

1. If the authors have adequately addressed your comments raised in a previous round of review and you feel that this manuscript is now acceptable for publication, you may indicate that here to bypass the “Comments to the Author” section, enter your conflict of interest statement in the “Confidential to Editor” section, and submit your "Accept" recommendation.

Reviewer #1: All comments have been addressed

2. Is the manuscript technically sound, and do the data support the conclusions?

Reviewer #1: Yes

3. Has the statistical analysis been performed appropriately and rigorously? 

Reviewer #1: Yes

4. Have the authors made all data underlying the findings in their manuscript fully available?

Reviewer #1: Yes

5. Is the manuscript presented in an intelligible fashion and written in standard English?

Reviewer #1: Yes

6. Review Comments to the Author

Reviewer #1: All concerns have been satisfactorily addressed. I appreciate the authors for their attendance to the details of the review comments.

7. PLOS authors have the option to publish the peer review history of their article (what does this mean?). If published, this will include your full peer review and any attached files.

Reviewer #1: No

---

## [Editor Report · Acceptance letter]

15 Nov 2022

PONE-D-22-13360R1 

Randomized Controlled Trial: Quantifying the Impact of Disclosing Uncertainty on Adherence to Hypothetical Health Recommendations 

Dear Dr. D'Agostino McGowan:

I'm pleased to inform you that your manuscript has been deemed suitable for publication in PLOS ONE. Congratulations! Your manuscript is now with our production department. 

Kind regards, 

on behalf of

Dr. Felix Bongomin 

Academic Editor

PLOS ONE